# Biomarker Analysis from a Phase I/Ib Study of Regorafenib and Nivolumab in Mismatch Repair-Proficient Advanced Refractory Colorectal Cancer

**DOI:** 10.3390/cancers16030556

**Published:** 2024-01-28

**Authors:** Dae Won Kim, Young-Chul Kim, Bence P. Kovari, Maria Martinez, Ruoyu Miao, James Yu, Rutika Mehta, Jonathan Strosberg, Iman Imanirad, Richard D. Kim

**Affiliations:** 1Department of Gastrointestinal Oncology, Moffitt Cancer Center, Tampa, FL 33612, USA; daewon.kim@moffitt.org (D.W.K.); maria.martinezjimenez@moffitt.org (M.M.); ruoyu.miao@moffitt.org (R.M.); james.yu@moffitt.org (J.Y.); rutika.mehta@moffitt.org (R.M.); jonathan.strosberg@moffitt.org (J.S.); iman.imanirad@moffitt.org (I.I.); 2Department of Biostatistics and Bioinformatics, Moffitt Cancer Center, Tampa, FL 33612, USA; youngchul.kim@moffitt.org; 3Department of Pathology, Moffitt Cancer Center, Tampa, FL 33612, USA; bence.kovari@moffitt.org

**Keywords:** nivolumab, regorafenib, colorectal cancer, mismatch repair-proficient, biomarker

## Abstract

**Simple Summary:**

Previous early phase studies demonstrated the modest but durable anticancer activity of nivolumab plus regorafenib in mismatch repair-proficient (pMMR) refractory colorectal cancer, suggesting the necessity of predictive biomarkers for better patient selection. The authors evaluated clinicopathological characteristics and the tumor microenvironment to identify potential biomarkers. Pretreatment albumin, MIP-1β, non-liver metastatic disease and regulatory T-cell infiltration may be potential predictive biomarkers of regorafenib/nivolumab in pMMR colorectal cancer.

**Abstract:**

Previously, we reported the modest but durable anticancer activity of regorafenib/nivolumab in mismatch repair-proficient (pMMR) refractory colorectal cancer in our I/Ib study. Our finding suggests the necessity of biomarkers for better selection of patients. Baseline clinical and pathological characteristics, blood and tumor samples from the patients in the trial were collected and evaluated to discover potential biomarkers. The obtained samples were assessed for immunohistochemistry, ELISA and RNA sequencing. Their correlations with clinical outcome were analyzed. A high albumin level was significantly associated with improved progression-free survival (PFS), overall survival (OS) and disease control. Non-liver metastatic disease showed prolonged PFS and OS. Low regulatory T-cell (Treg) infiltration correlated with prolonged PFS. Low MIP-1β was associated with durable response and improved OS significantly. Upregulation of 23 genes, including CAPN9, NAPSA and ROS1, was observed in the durable disease control group, and upregulation of 10 genes, including MRPS18A, MAIP1 and CMTR2, was associated with a statistically significant improvement of PFS. This study suggests that pretreatment albumin, MIP-1β, non-liver metastatic disease and Treg infiltration may be potential predictive biomarkers of regorafenib/nivolumab in pMMR colorectal cancer. Further studies are needed to confirm these findings.

## 1. Introduction

Due to the remarkable success of immune checkpoint inhibitors in diverse cancers, immunotherapy has emerged as another pillar of cancer therapy in addition to surgery, radiation and chemotherapy. Programmed cell death 1 (PD-1) and its ligand, PD-L1 blockade immunotherapy, have demonstrated significant anticancer activity in treatment-naïve and refractory microsatellite instability-high (MSI-H)/mismatch repair-deficienct (dMMR) metastatic colorectal cancers, which represent less than 5% of metastatic diseases [1,2]. However, PD-1/PD-L1 blockade immunotherapy failed to show any significant clinical benefit in microsatellite-stable (MSS)/MMR-proficient (pMMR) metastatic colorectal cancer [3,4]. The combination of an antiPD-1 or antiPD-L1 inhibitor and other therapeutic agents has been extensively studied to improve the clinical outcome of refractory MSS/pMMR colorectal cancer, and we reported a phase I/Ib study of nivolumab plus regorafenib in patients with refractory pMMR colorectal cancer recently [5]. Although our data showed limited anticancer activity of nivolumab plus regorafenib, a durable response was observed in selected patients, suggesting the necessity of predictive biomarkers for precise patient selection and improvement of clinical outcomes. In this study, baseline clinical and pathological characteristics, pretreatment blood and pretreatment tumor samples were analyzed to identify potential biomarkers of nivolumab plus regorafenib in pMMR colorectal cancer patients.

## 2. Materials and Methods

### 2.1. Study Population

Baseline clinical and pathological characteristics and pretreatment blood and tumor samples were collected from patients with refractory pMMR colorectal cancer who were enrolled in the phase I/Ib study of nivolumab plus regorafenib (ClinicalTrials.gov (accessed on 23 May 2023), identifier: NCT 03712943). After approval from the local Institutional Review Boards and written informed consent from patients, enrolled patients received nivolumab 240 mg intravenously every 2 weeks and regorafenib 80 mg or 120 mg daily for 3 weeks on/1 week off orally until disease progression or unacceptable toxicity, as previously reported [6]. A total of 51 patients received at least one dose of treatment.

### 2.2. Immunohistochemical Staining

CD4, CD8, CD68, CD163, FOXP3, PD-1 and PD-L1 expression in tumor microenvironment was evaluated in pretreatment tumor samples. The analysis was conducted as previously reported [6]. Briefly, tissue sections were incubated with each antibody before incubation with the secondary antibody. The tissue sections were incubated with streptavidin–biotin complex after washing the secondary antibody. Stained tumor samples were evaluated by two GI pathologists after confirmation of positive and negative controls. Positive result was defined if ≥1% of tumor cells were stained with anti-PD-L1 antibody. Any lymphocytes stained with anti-CD4, anti-CD8, anti-CD68, anti-CD163, anti-PD-1 and anti-FOXP3 were considered as positive. Regulatory T cell (Treg) was defined by CD4- and FOXP3-expressing T cells.

### 2.3. Cytokine/Chemokine Analysis

Pretreatment serum samples were collected for cytokine and chemokine analysis. The serum levels of 16 biomarkers, including IL-2, IL-4, IL-5, IL-6, IL-7, IL-8, IL-13, IL-15, IL-17A, IL-18, IP-10, MCP-1, MIP-1α, MIP-1β, TNF-α and VEGF-A, were measured with U-PLEX multiplex assay platform (Meso Scale Diagnostics, Rockville, MD, USA) as described previously [7].

### 2.4. RNA Sequencing

Extracted RNA samples from the tumor samples were processed for RNA sequencing (RNAseq) as previously reported [6]. Briefly, GI pathologists reviewed H & E-stained tumor slides to select and annotate tumor area. Histotechnologists cut thick sections (10–20 µm) onto slides and overlayed the annotated H & E slides over the unstained slide to scrape away the non-tumor tissue. The selected tumor portion was used for RNA extraction using RNeasy FFPE Kit (Qiagen, Germantown, MD, USA). Using TruSeq RNA Exome kit (Illumina, San Diego, CA, USA), RNA was bi-directionally sequenced in a 2 × 100 paired-end configuration on the Illumina NextSeq 2000 sequencer (Illumina). The adjusted *p*-value of 0.0001 was used as cutoff.

### 2.5. Statistical Methods

Fisher’s exact test was used to assess the association between categorical variables, including baseline albumin level (high versus low), and clinical response such as overall survival, progression-free survival, objective response and disease control as previously reported [6]. The cutoff value of high albumin level was >3.7 g/dL based on previously reported data [8]. Objective response included complete response and partial response, and disease control included objective response and stable disease based on Response Evaluation Criteria in Solid Tumors (RECIST 1.1). The association between expression of CD4/FOXP3, CD8, CD68, CD163, PD-1 or PD-L1 and clinical outcome was evaluated using Kaplan–Meier survival and log-rank test. Cox proportional hazard regression was used for multivariate analysis, and hazard ratio (HR) and 95% confidence interval were estimated. All statistical analyses were performed using R Version 4.2.1. All statistical tests used a significance level of 5%. No adjustments for multiple testing were made if not specified otherwise.

### 2.6. RNAseq Analysis

The analysis was conducted as previously reported [6]. Briefly, differentially expressed genes from RNAseq were evaluated using the R DESeq2 package (Version 1.42.0) and a false discovery rate (FDR) of <0.05 and absolute fold change cutoff of >2. The association of gene expression with durable disease control and rapid disease progression was analyzed using Cox proportional hazards regression with a nominal *p*-value cutoff of <0.01 and absolute hazard ratio of >2. Durable disease control was defined as a disease that was at least stable over 6 months, and rapid disease progression was defined as disease progression within 2 months. Pathway enrichment and annotation analyses of differentially expressed genes were completed using the R EnrichR package (Version 3.2) based on Kyoto Encyclopedia of Genes and Genomes (KEGG) database.

## 3. Results

### 3.1. Baseline Clinicopathological Characteristics and Clinical Outcome

Baseline clinicopathological characteristics, progression-free survival (PFS) and overall survival (OS) of 51 patients who were enrolled in the phase I/Ib study of nivolumab plus regorafenib [5] are summarized in Table 1. High pretreatment serum albumin level was associated with prolonged PFS (median: 5.6 months versus 1.9 months, HR 0.34, 95% confidence interval (CI): 0.16–0.72, *p* = 0.025) and OS (median: 15.5 months versus 3.1 months, HR 0.20, 95% CI: 0.10–0.37, *p* = 0.0001). Metastatic disease involving non-liver sites showed improved PFS (median: 11.7 months versus 2.3 months, HR 0.37, 95% CI: 0.19–0.73, *p* = 0.006) and OS (median: 19.5 months versus 9.1 months, HR 0.40, 95% CI: 0.22–0.74, *p* = 0.01) compared with metastasis involving the liver (Table 1). Interestingly, metastatic disease involving non-lung sites was associated with poor PFS (median: 2.3 months versus 5.6 months, HR 2.3, 95% CI: 1.07–4.76, *p* = 0.009) compared with metastasis involving the lungs (Table 1). We did not observe any other correlation between baseline characteristics and PFS or OS (Table 1). The correlation between albumin level and the presence or absence of metastatic liver disease was evaluated since albumin level can be affected by liver function. The low albumin level was observed in 13 patients (35%) with liver metastatic disease and 1 (7%) with non-liver metastasis. However, it was not statistically significant (*p* = 0.077) (Appendix A).

Baseline patient characteristics and corresponding tumor response, including objective response rate (ORR) and disease control rate (DCR), were analyzed in 40 evaluable patients defined by completion of the first set of scans (Table 2). All responders were white, and they had high serum albumin levels (Table 2). High albumin correlates with improved DCR (71.9% versus 25%, *p* = 0.01), and metastatic disease involving non-liver sites was associated with high ORR (25% versus 3.6%, *p* = 0.04). No other correlation was observed (Table 2).

### 3.2. Immunohistochemical Staining

Baseline expression of CD8, CD68, CD163, PD-1, PD-L1 and regulatory T cells (Treg) in the tumor microenvironment (Appendix A) was evaluated with clinical outcome. Among 23 available pretreatment tumor samples, 17 patients’ samples with liver metastasis were collected in the liver, and 6 patients’ samples with non-liver metastasis were collected in the lung (*n* = 4), lymph node (*n* = 1) and peritoneum (*n* = 1) within 1 week prior to the cycle 1 dose of nivolumab and regorafenib. We observed PD-1 expression in 8 patients’ samples and PD-L1 expression in 11 patients’ samples. DCR of PD-1-expressing patients was 75% with one partial response, while DCR of PD-L1-expressing patients was 36.4% with one partial response. No correlation between the expression of PD-1 or PD-L1 and PFS or OS was observed (Appendix A).

A high frequency of Treg in the tumor microenvironment is associated with short PFS (median: 1.9 months versus 9.7 months, HR 5.11, 95% CI: 1.82–14.34, *p* = 0.006) compared with low Treg (Figure 1). We did not observe any prognostic role of baseline CD8, CD68 or CD163 expression in the tumor microenvironment (Figure 1).

We evaluated the baseline density of CD8 T cells, Tregs and macrophages (CD68 and CD163) with the metastatic sites (liver and non-liver sites). High densities of the Treg and CD163 macrophages were observed in 8 patients (47%) and 9 patients (53%) with the liver metastatic tumor microenvironment compared with 1 (17%) and 1 (17%) in the non-liver metastatic tumor microenvironment. However, it was not statistically significant (Appendix A).

### 3.3. Cytokine/Chemokine Analysis

Baseline levels of 16 cytokine and chemokine biomarkers were measured in the serum of 27 patients (Appendix A). To identify potential prognostic markers, we investigated the association between baseline levels of all biomarkers and the clinical outcome, including PFS, OS and durable disease control, defined as at least stable disease over 4 months. A low baseline level of MIP-1β is associated with durable disease control (mean level: 283 pg/mL for durable disease versus 408 pg/mL for non-response, *p* = 0.02) and prolonged OS (median: 18 months versus 7.7 months, HR 0.43, 95% CI 0.19–0.96, *p* = 0.012) compared with a high level of MIP-1β (Figure 2). We observed trending between prolonged OS and a low baseline level of IL-6 (median: 14.3 months versus 8.8 months, *p* = 0.051) or a low baseline level of MIP-1α (median: 13.1 months versus 8.9 months, *p* = 0.055) without statistical significance (Appendix A).

The correlation between MIP-1β and liver metastasis or macrophages (CD68 and CD163) was evaluated, and we observed high baseline MIP-1β levels (mean: 375 pg/mL versus 300 pg/mL, *p* = 0.2) in patients with liver metastasis compared with non-liver metastasis without any statistical significance. No correlation between MIP-1β and the density of macrophages in the tumor microenvironment was observed.

### 3.4. RNAseq Analysis

To discover potential biomarkers, pretreatment tumor samples from 10 patients with durable disease control and 11 patients with rapid disease progression were collected for RNAseq analysis. Durable disease control was defined as a disease that was at least stable over 6 months, and rapid disease progression was defined as disease progression within 2 months. Upregulation of 23 genes, including CAPN9, NAPSA, ROS1, CDHR1, SFTPA2, NXPE1, AGBL1, VWA5B1, CRTAC1 and VWA3B, and downregulation of 17 genes, including ADIPOQ, AL138789.1, LYPD3, PADI1, NTSR1, PGAM1P7, ARHGEF4, KRTAP3-1, CLDN11 and DCBLD2, were observed in the durable disease control group with statistical significance (FDR < 5%) compared with the rapid disease progression group (Figure 3 and Appendix A). We observed that upregulation of certain genes, including MRPS18A, MAIP1, CMTR2, TSKU, SCARNA18B, AC005632.2, AC003002.2, DIP2B, TOLLIP and HADHA, was associated with a statistically significant improvement of PFS, while upregulation of C7orf69, ZNF738, HNRNPCL1, GPRIN3, AC034236.1, ZNF436-AS1, AC027644.4, GPATCH2, RIC1 and VEGFC was associated with decreased PFS (Appendix A). Hallmark pathway analysis and KEGG enrichment analysis were performed to identify the potential pathways affected by differentially expressed genes in each group. Downregulated gene expression involving apical surface pathways and oxidative phosphorylation pathways was associated with good disease control (FDR *p* = 0.006) and prolonged PFS (FDR *p* = 0.006), respectively.

## 4. Discussion

Although immune checkpoint inhibitors demonstrated remarkable anticancer activity in MSI-H/dMMR colorectal cancer [9], they failed to show any clinical benefit in MSS/pMMR colorectal cancer, which is over 90% of colorectal cancer. Extensive efforts have been made to improve the anticancer activity of immune checkpoint inhibitors in MSS/pMMR colorectal cancer by conversion of an immunosuppressive “cold tumor” to an immunogenic “hot tumor”. Previously, we reported a moderate but durable response to nivolumab plus regorafenib in unselected patients with refractory MSS/pMMR colorectal cancer [5], suggesting identifying predictive biomarkers can improve clinical outcomes by precise patient selection. In this study, baseline clinical and pathological characteristics and pretreatment blood/tumor samples were investigated to discover potential biomarkers of nivolumab plus regorafenib in patients with pMMR colorectal cancer in our previous phase 1/1b trial [5]. A high baseline albumin level was associated with a high disease control rate (71.9% versus 25%, *p* = 0.01), prolonged PFS (5.6 months versus 1.9 months, *p* = 0.025) and OS (15.5 months versus 3.1 months, *p* = 0.0001). This finding is consistent with previous reports suggesting pretreatment albumin can be a potential biomarker predicting a favorable clinical outcome following PD-1 blockade immunotherapy in advanced solid cancers, including gastric cancer, hepatocellular carcinoma, lung cancer, ovarian cancer and urothelial cancer [8,10]. This finding is explained by the fact that malnutrition and chronic inflammation decrease albumin synthesis [11], and malnutrition and chronic inflammation have been reported as potential poor prognostic markers of immune checkpoint inhibitor therapy [12,13]. Since albumin level can be affected by liver function, we evaluated the correlation between the baseline albumin level and the presence of metastatic liver disease. The low albumin level (35.1% in liver metastasis versus 7.1% in non-liver metastasis) was observed in more patients with liver metastatic disease than with non-liver metastasis. However, it was not statistically significant (*p* = 0.077) (Appendix A).

We observed a significantly improved clinical outcome of metastatic disease involving non-liver sites compared with metastatic disease involving the liver, and this finding is consistent with previous reports [14,15]. Previous pre-clinical data suggested that elimination of tumor-specific T cells by immunosuppressive hepatic macrophages [16] and activation of Tregs [17] may be potential mechanisms of a diminished response to immunotherapy in liver metastatic disease. We observed a high density of Tregs (47.1% in liver metastasis versus 16.7% in non-liver metastasis, *p* = 0.2) and CD163 macrophages (52.9% in liver metastasis versus 16.7% in non-liver metastasis, *p* = 0.1) in the liver metastatic microenvironment compared with non-liver metastatic sites, although it has no statistical significance partly due to the small sample size (Appendix A). Given the small sample size in our study, further evaluation with more samples may elucidate the mechanisms of the poor response to immunotherapy in liver metastatic disease.

While the low frequency of Tregs was associated with improved PFS in the tumor microenvironment (Figure 1), no predictive value of CD8, CD68, CD163, PD-1 and PD-L1 expression was observed in our study. The potential predictive role of these markers is very controversial in colorectal cancer. While previous data demonstrated no predictive/prognostic role of the expression of CD8 or PD-L1 in patients with metastatic MSS/pMMR colorectal cancer treated with regorafenib plus avelumab [18], another study suggested CD8 and PD-L1 might be potential biomarkers in MSS/pMMR colorectal cancer treated with regorafenib plus nivolumab [15]. These two studies [15,18] used regorafenib to induce the immunogenic tumor microenvironment of MSS/pMMR colorectal cancer, similar to our study. Tregs and macrophages (CD68 and CD163) are more complicated since some data suggested Treg and macrophage infiltration in the tumor microenvironment might be poor prognostic markers [18,19], but another study suggested they might be favorable prognostic markers [15,20]. These discrepancies are likely from the difficulties in identifying and characterizing immune-suppressive cells. FOXP3 is commonly used as a Treg marker [21]. However, FOXP3-expressing T cells are heterogenous in colorectal cancer [19], and certain types of FOXP3-expressing T cells (FOXP3lo) and activation-induced FOXP3-expressing T cells have been reported as non-immunosuppressive T cells [19,22]. Further studies for better identification and characterization of immunosuppressive cells with a functional assay may be needed to clarify the discrepancies in colorectal cancer. In addition, cautious interpretation is needed for biomarker analysis in colorectal cancer since the biology and tumor microenvironment are quite different to MSI-H/dMMR from MSS/pMMR, and therapeutic agents combined with immunotherapy can induce different tumor microenvironment due to a different mechanism of action of each therapeutic agent.

Our data demonstrated high serum levels of MIP-1β were associated with poor response (mean of MIP-1β: 283 pg/mL for durable response versus 408 pg/mL for non-response, *p* = 0.02) and reduced survival time (median OS: 7.7 months versus 18 months, *p* = 0.012) after nivolumab plus regorafenib (Figure 2). MIP-1β (CCL4) is a chemoattractant for immune regulatory cells, and it was suggested as a potential immune-related prognostic biomarker [23]. MIP-1β was reported as a poor prognostic marker with a positive correlation between MIP-1β and immune-suppressive pro-tumor macrophage infiltration in colorectal cancer [24]. In addition, MIP-1β has been suggested as a potential biomarker to predict the immunotherapy response in renal cell carcinoma [25]. Since MIP-1β is produced by macrophages, the correlation between MIP-1β and the density of macrophages (CD68 and CD163) in the tumor microenvironment was evaluated. However, no correlation was observed between MIP-1β and the density of macrophages in the tumor microenvironment. Our data suggest further evaluation is needed to define the predictive and prognostic role of MIP-1β in colorectal cancer.

We observed differentially expressed genes between the durable disease control and rapid disease progression groups with statistical significance. Among the 23 significantly upregulated genes in the durable disease control group, SFTPA1 was reported as a potential predictive biomarker for immunotherapy for lung adenocarcinoma [26]. Among the 17 significantly downregulated genes in the durable disease control group, DCBLD2 was reported as a potential poor prognostic marker in colorectal cancer and lung cancer by stimulation of epithelial-to-mesenchymal transition [27,28]. In addition, DCBLD2 was associated with a poor response to PD-1 blockade immunotherapy in patients with melanoma and bladder cancer [29].

Downregulated gene expression involving an oxidative phosphorylation pathway was associated with prolonged PFS (FDR *p* = 0.006), and this finding is consistent with previous data demonstrating upregulation of oxidative phosphorylation-related genes can be one of the resistance mechanisms of checkpoint inhibitor immunotherapy in melanoma [30]. Fakih et al. reported that upregulation of certain genes related to immune sensitivity, including allograft rejection, interferon γ response, IL-6/JAK/STAT signaling and interferon α response, was associated with an improved clinical outcome in pMMR colorectal cancer treated with nivolumab plus regorafenib [15]. However, we did not observe it in our study, likely due to the small sample size of our study (*n* = 21) and their study (*n* = 27).

Our major limitations are that this study is a single-arm study without control groups, and the sample size of this study is relatively small with insufficient pretreatment tumor samples, which may cause selection bias.

## 5. Conclusions

Our study suggests high pretreatment albumin levels, low pretreatment MIP-1β levels, non-liver metastasis, low Treg density and different gene expression in the tumor microenvironment may be potential predictive biomarkers of nivolumab plus regorafenib in refractory pMMR colorectal cancer. Further large-scale studies are warranted to verify our findings for improved treatment selection for patients with pMMR colorectal cancer.

## Figures and Tables

**Figure 1 cancers-16-00556-f001:**
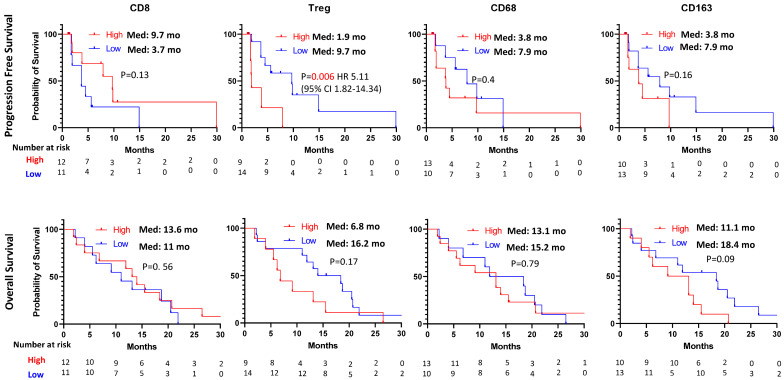
Kaplan–Meier estimates of progression-free survival and overall survival by CD8, Treg (CD4/FOXP3), CD68 and CD163.

**Figure 2 cancers-16-00556-f002:**
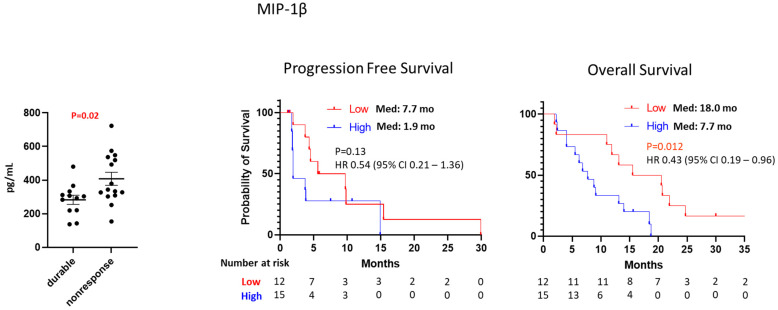
Baseline MIP-1β level between durable responders and non-responders. Kaplan–Meier estimates of progression-free survival and overall survival by MIP-1β.

**Figure 3 cancers-16-00556-f003:**
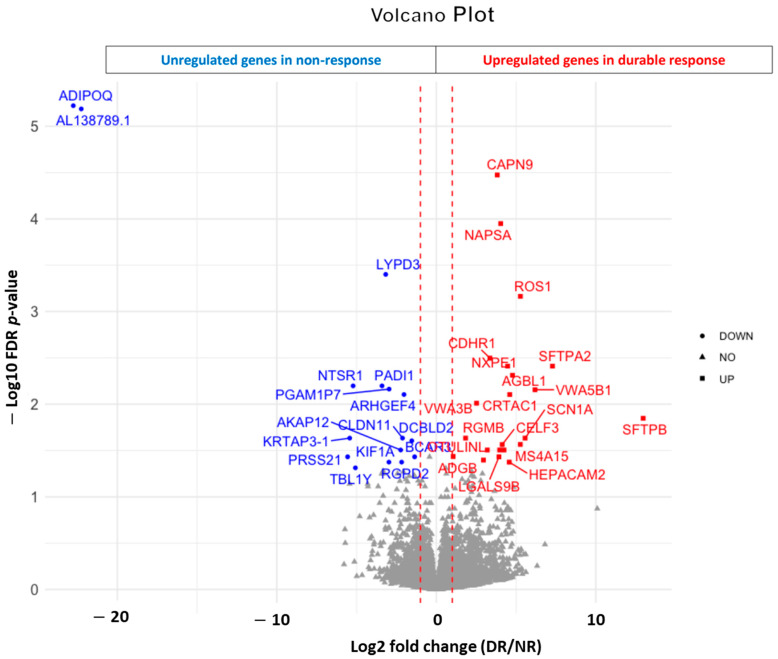
Volcano plot of differentially expressed gene expression profiles associated with durable response.

**Table 1 cancers-16-00556-t001:** Baseline characteristics and corresponding survival outcome.

	*n* = 51	Median PFS (Months)		Median OS (Months)	
Age					
Median (range)	56 (31–79)	4.3	95% CI 1.6–7.0	11.1	95% CI 8.1–14.1
Gender					
Male	27 (53.8%)	3.8	HR 1.04 (95% CI 0.53–2.05)	9.7	HR 1.17 (95% CI 0.65–2.10)
Female	24 (46.2%)	4.3	*p* = 0.9	14.3	*p* = 0.61
Race					
White	42 (80.8%)	4.3	*p* = 0.25	11.5	*p* = 0.36
Hispanic	3 (5.8%)	not reached		10	
African	2 (5.8%)	3		20.9	
Asian	4 (7.7%)	3.8		19	
Obesity					
Yes	15 (29.4%)	3.7	HR 0.7 (95% CI 0.35–1.40)	18.4	HR 0.72 (95% CI 0.39–1.35)
No	36 (70.6%)	4.3	*p* = 0.32	10.5	*p* = 0.33
Albumin					
High	37 (71.2%)	5.6	HR 0.34 (95% CI 0.16–0.72)	15.5	HR 0.20 (95% CI 0.10–0.37)
Low	14 (28.8%)	1.9	* p * = 0.025	3.1	* p * = 0.0001
ECOG PS					
0	20 (38.5%)	4	HR 1.28 (95% CI 0.66–2.45)	11.6	HR 0.92 (95% CI 0.51–1.68)
1	31 (61.5%)	5.6	*p* = 0.6	7.9	*p* = 0.068
Primary tumor					
Left-sided	21 (42.3%)	3.8	HR 1.15 (95% CI 0.57–2.28)	14	HR 0.65 (95% CI 0.36–1.17)
Right-sided	30 (57.7%)	4.3	*p* = 0.7	10.5	*p* = 0.16
Previous systemic treatment					
2nd line	30 (57.7%)	5.6	HR 0.72 (95% CI 0.37–1.40)	13.1	HR 1.0 (95% CI 0.55–1.83)
≥3rd line	21 (42.3%)	3	*p* = 0.31	11	*p* = 0.9
RAS mutation status					
WT	14 (28.8%)	2.9	HR 1.35 (95% CI 0.66–2.77)	11	HR 1.03 (95% CI 0.54–1.97)
Mutant	37 (71.2%)	5.6	*p* = 0.38	11.9	*p* = 0.9
Sites of disease					
Liver involvement					
No	14 (27.5%)	11.7	HR 0.37 (95% CI 0.19–0.73)	19.5	HR 0.40 (95% CI 0.22–0.74)
Yes	37 (72.5%)	2.3	* p * = 0.006	9.1	* p * = 0.01
Lung involvement					
No	19 (37.3%)	2.3	HR 2.3 (95% CI 1.07–4.76)	10	HR 1.03 (95% CI 0.56–1.91)
Yes	32 (62.7%)	5.6	* p * = 0.009	12.1	*p* = 0.9
Peritoneal involvement					
No	41 (80.4%)	4.5	HR 0.97 (95% CI 0.34–2.79)	13.1	HR 0.54 (95% CI 0.22–1.33)
Yes	10 (19.6%)	1.9	*p* = 0.9	5.9	*p* = 0.09

CI: confidence interval; HR: hazard ratio; PFS: progression-free survival.

**Table 2 cancers-16-00556-t002:** Baseline clinicopathological characteristics and corresponding tumor response.

	*n* = 40 (Evaluable Patients)	PR (*n* = 4)	SD (*n* = 21)	ORR	*p* Value	DCR	*p* Value
Age							
Median (range)	56 (31–78)	75 (54–78)	57 (31–70)				
Gender							
Male	24	1 (4.2%)	13 (54.2%)	4.2%	*p* = 0.2	58.4%	*p* = 0.9
Female	16	3 (18.8%)	8 (50.0%)	18.8%		68.8%	
Race							
White	34	4 (11.8%)	18 (52.9%)	11.8%	*p* = 0.6	64.7%	*p* = 0.7
Hispanic	0	0	0	0		0	
African	2	0	1 (50.0%)	0		50.0%	
Asian	4	0	2 (50.0%)	0		50.0%	
Obesity							
Yes	13	2 (15.4%)	6 (46.2%)	15.4%	*p* = 0.4	61.5%	*p* = 0.9
No	27	2 (7.4%)	15 (55.6%)	7.4%		63.0%	
Albumin							
High	32	4 (12.5%)	19 (59.4%)	12.5%	*p* = 0.2	71.9%	* p * = 0.01
Low	8	0	2 (25%)	0		25.0%	
ECOG PS							
0	18	1 (5.6%)	9 (50%)	5.6%	*p* = 0.4	55.6%	*p* = 0.4
1	22	3 (13.6%)	12 (54.5%)	13.6%		68.2%	
Primary tumor							
Left-sided	18	3 (16.7%)	9 (37.5%)	16.7%	*p* = 0.2	66.7%	*p* = 0.6
Right-sided	22	1 (4.5%)	12 (54.5%)	4.5%		59.1%	
Previous systemic treatment							
2nd line	21	3 (14.3%)	12 (57.1%)	14.3%	*p* = 0.3	71.4%	*p* = 0.2
≥3rd line	19	1 (5.3%)	9 (47.3%)	5.3%		52.6%	
RAS mutation status							
WT	13	2 (15.4%)	5 (38.5%)	15.4%	*p* = 0.4	53.9%	*p* = 0.4
Mutant	27	2 (7.4%)	16 (59.3%)	7.4%		66.7%	
Sites of disease							
Liver involvement							
No	12	3 (25.0%)	7 (58.3%)	25%	* p * = 0.04	83.3%	*p* = 0.07
Yes	28	1 (3.6%)	14 (50.0%)	3.6%		53.6%	
Lung involvement							
No	15	1 (6.7%)	6 (40.0%)	6.7%	*p* = 0.6	46.7%	*p* = 0.1
Yes	25	3 (12.0%)	15 (60.0%)	12.0%		72.0%	
Peritoneal involvement							
No	36	3 (8.3%)	21 (58.3%)	8.3%	*p* = 0.3	66.7%	*p* = 0.1
Yes	4	1 (25.0%)	0	25.0%		25.0%	

CI: confidence interval; DCR: disease control rate; HR: hazard ratio; ORR: objective response rate; PR: partial response; SD: stable disease.

## Data Availability

All authors had access to the data published in this paper. An anonymized dataset may be available from the corresponding author on reasonable request.

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
