# Peer review of "Biomarker Analysis from a Phase I/Ib Study of Regorafenib and Nivolumab in Mismatch Repair-Proficient Advanced Refractory Colorectal Cancer"

_cancers, 2024, doi:10.3390/cancers16030556_

Round 1

Reviewer 1 Report

Comments and Suggestions for Authors

Kim et al are reporting the biomarker analysis from a phase 1/1b rego+nivo trial in chemorefractory CRC. The author are reporting data and interpretation from the phase 1 trial, from which 23 pretreatment tumor samples were analyzed by IHC, 27 blood samples for analysis, and 21 for RNAseq. The authors should be commended for a fairly detailed analysis of many relevant immune biomarkers, as well as RNAseq and pathways analysis. The figures and tables are logical and easy to interpret. There are some interesting correlations uncovered here, though as the authors rightly point out, the impact will be limited due to the small sample size. While most of the findings are not necessarily brand new findings, the MIP correlation w/ a clinical outcome is novel. Since this is a chemokine with immune function, it is highly relevant to the treatment/study, and well worth further investigation. The manuscript is well written. I would suggest the following edits to enhance the scientific rigor of the manuscript.

1. Page 7 of 12, lines 191-193. 23 tumor specimens were analyzed for much of the tissue studies.  However, 11+10 = 21 specimens were analyzed for RNAseq.  What happened to the other 2?

2. Page 3 of 12, lines 108-114. RNAseq technique: Were tumor areas selected based on histology, and dissected from slides to enrich for tumor DNA? Or were whole sections used for RNA extraction? Was the sequencing done w/ bi-directional or uni-directional sequencing?

3. Discussion, page 8 of 12, lines 251-260.  I think the discussion point about prognostic value of these immune markers should be given some additional context here.  The authors reference 3 studies with conflicting findings about the value of these markers (refs 3, 13, 16). However, they should point out that one of these studies was in a MSI-H population, and two are in MSS CRC. Furthermore, the MSS studies are in immune combinations, where there was a hypothesis for immunomodulation by a TKI.  I think it is important to include this context because it further muddies the ability to interpret the biomarker data. Also, the MSI versus MSS distinction is key. I consider these to be 2 distinct diseases.

Author Response

  1. Page 7 of 12, lines 191-193. 23 tumor specimens were analyzed for much of the tissue studies.  However, 11+10 = 21 specimens were analyzed for RNAseq.  What happened to the other 2?

We appreciate your question. 23 tumor samples were used for immunohistochemical staining first then RNAseq with left over samples. Unfortunately, we could not obtain enough RNA extraction from 2 samples.

  1. Page 3 of 12, lines 108-114. RNAseq technique: Were tumor areas selected based on histology, and dissected from slides to enrich for tumor DNA? Or were whole sections used for RNA extraction? Was the sequencing done w/ bi-directional or uni-directional sequencing?

We appreciate your comment. Our GI pathologists reviewed H & E stained tumor slides to annotate tumor area. Histotechnologists cut thick sections onto slides, and overlayed the annotated H & E slides over the unstained slide to scarp away the non-tumor tissue. The selected tumor portion was used for RNA extraction. It was bi-directional sequencing. This was added in Methods section. 

  1. Discussion, page 8 of 12, lines 251-260.  I think the discussion point about prognostic value of these immune markers should be given some additional context here.  The authors reference 3 studies with conflicting findings about the value of these markers (refs 3, 13, 16). However, they should point out that one of these studies was in a MSI-H population, and two are in MSS CRC. Furthermore, the MSS studies are in immune combinations, where there was a hypothesis for immunomodulation by a TKI.  I think it is important to include this context because it further muddies the ability to interpret the biomarker data. Also, the MSI versus MSS distinction is key. I consider these to be 2 distinct diseases.

We appreciate your suggestion and we agree with you. Cautious interpretation is needed for biomarker analysis in colorectal cancer since biology and tumor microenvironment is quite different of MSI-H/dMMR from MSS/pMMR, and different therapeutic agents combined immunotherapy can change tumor microenvironment with different mechanism of action.  We added this point in the discussion section.   

Reviewer 2 Report

Comments and Suggestions for Authors

The manuscript "Biomarker Analysis from a Phase I/Ib Regorafenib and Nivolumab in Mismatch Repair Proficient Advanced Refractory  Colorectal Cancer" by 

 Dae Won Kim is an exciting paper that contributes to the advances in precision medicine.  

The treatment of metastatic colorectal cancer is challenging. Selecting appropriate therapeutic agents is crucial to extend the progression-free survival and overall survival rates.

The treatment with a combination of an antiPD-1 or antiPD-L1 inhibitor or other therapeutic agents is effective in naïve and refractory microsatellite instability-high (MSI-H)/mismatch repair deficiency (dMMR) metastatic colorectal cancers, which represent less than 5 % of metastatic disease. This type of treatment has limited anticancer activity in refractory MSS/pMMR metastatic colorectal cancer, suggesting the necessity of predictive biomarkers for patient selection and improvement of clinical outcomes. 

This study analyzed potential biomarkers for patients with Mismatch Repair Proficient Advanced Refractory Colorectal Cancer selection to treatment with Regorafenib and Nivolumab. The authors suggest that pretreatment albumin, MIP-1β, non-liver metastatic disease, and Tregs infiltration are the biomarkers that can help in patient selection. This way, using these biomarkers in patient selection can maximize the therapeutic results, minimize costs, and improve patients' survival and quality of life.

The study design is well executed, the statistical analysis appears appropriate, and the results are well presented. The discussion is satisfactory. The study limitations are written in the manuscript: "Our major limitations are that this study is a single arm study without control groups, and the sample size of this study is relatively small samples size with insufficient pretreatment tumor samples which may cause selection bias."The conclusions are adequate. The references are suitable.

I suggest minor alterations:

"The cutoff value of high albumin level was > 3.7 g/dL."

"Objective response rate (ORR) and disease control rate (DCR) definition"

“Treg was defined 149 by CD4 and FOXP3 expressing T cells”

"Disease control was defined as at least stable disease over six months, and rapid disease progression was defined as disease progression within two months."

The periodicity and exams executed to define "disease control" and "disease progression."

Should be included in "Materials and Methods" and not in "Results."

Author Response

I suggest minor alterations:

"The cutoff value of high albumin level was > 3.7 g/dL."

"Objective response rate (ORR) and disease control rate (DCR) definition"

“Treg was defined 149 by CD4 and FOXP3 expressing T cells”

"Disease control was defined as at least stable disease over six months, and rapid disease progression was defined as disease progression within two months."

The periodicity and exams executed to define "disease control" and "disease progression."

Should be included in "Materials and Methods" and not in "Results."

We appreciate your suggestion. All these information was described in Methods section.

Reviewer 3 Report

Comments and Suggestions for Authors

Dear Authors

Clinical study describes pretreatment high albumin level, pretreatment low MIP-1β level, non-liver metastasis, low Treg density and different gene expression in tumor microenvironment may be potential predictive biomarkers of nivolumab plus regorafenib in refractory pMMR colorectal cancer. Main limitations are that this study is a single arm study without control groups, and the sample size of this study is relatively small.

The following step should provide clear information for readers to enjoy it,

1)      Materials and Methods - Study population section – please mention how many patients are involved in this study?

2)      Immunohistochemical staining - CD4, CD8, CD68, CD163, FOXP3, PD-1 and PD-L1 expression in pretreatment tumor samples – please keep figures in the main text.

3)      RNA sequencing – cDNA – mention kit name, catalog number.

Comments on the Quality of English Language

Moderate editing of English language required.

Author Response

The following step should provide clear information for readers to enjoy it,

  1. Materials and Methods - Study population section – please mention how many patients are involved in this study?

We appreciate your question. A total 51 patients were received at least one dose of treatment and it was added in the study population section.

2)      Immunohistochemical staining - CD4, CD8, CD68, CD163, FOXP3, PD-1 and PD-L1 expression in pretreatment tumor samples – please keep figures in the main text.

We appreciate your suggestion. Representative figures of immunohistochemical staining were added in supplemental data (Supplement figure 1).

3)      RNA sequencing – cDNA – mention kit name, catalog number.

We appreciate your comment. We used RNeasy FFPE Kit (quagen cat number 73504) for RNA extraction and TruSeq RNA Exome kit (Illumina PN 20020189) for RNA seq. It was added in Methods section.